# The Value of the Naples Prognostic Score at Diagnosis as a Predictor of Cervical Cancer Progression

**DOI:** 10.3390/medicina61081381

**Published:** 2025-07-30

**Authors:** Seon-Mi Lee, Hyunkyoung Seo, Seongmin Kim, Hyun-Woong Cho, Kyung-Jin Min, Sanghoon Lee, Jin-Hwa Hong, Jae-Yun Song, Jae-Kwan Lee, Nak-Woo Lee

**Affiliations:** 1Department of Obstetrics and Gynecology, Korea University College of Medicine, 73 Koreadae-ro, Seongbuk-gu, Seoul 02841, Republic of Korea; tjsal4142@naver.com (S.-M.L.);; 2Department of Obstetrics and Gynecology, Korea University College of Medicine, 148 Gurodong-ro, Guro-gu, Seoul 08308, Republic of Korea; 3Department of Obstetrics and Gynecology, Korea University College of Medicine, 123 Jeokgeum-ro, Dannon gu, Ansan-si 15355, Gyeonggi-do, Republic of Korea

**Keywords:** cervical cancer, gynecological cancer, Naples prognostic score, prognostic factor, cancer progression

## Abstract

*Background and Objectives*: The Naples prognostic score (NPS), which incorporates inflammatory and nutritional indicators, is increasingly used as a prognostic score for various malignancies. Nonetheless, few studies have specifically evaluated the NPS as a prognostic factor for cervical cancer. This study aimed to assess the value of NPS at diagnosis as a predictor of cancer progression. *Materials and Methods*: This study included patients diagnosed with cervical cancer at Korea University Anam Hospital from January 2019 to December 2023. Patients with incomplete data or those who were lost to follow-up were excluded. The NPS was calculated based on laboratory results at the time of diagnosis, categorizing patients into the low-NPS group (NPS 0–1) and high-NPS group (NPS ≥ 2). Survival analysis was performed using the Kaplan–Meier method and log-rank test. Univariate and multivariate Cox proportional hazards models were used to identify independent prognostic factors. *Results*: Out of 178 patients, 98 and 80 were categorized into the low-NPS and high-NPS groups, respectively. Kaplan–Meier survival analysis showed that the high-NPS group had significantly lower disease-free survival (DFS) (*p* < 0.001) and overall survival (OS) (*p* = 0.02) rates than the low-NPS group. Multivariate Cox regression analysis identified the NPS as an independent prognostic factor for DFS (adjusted hazard ratio, 1.98; *p* = 0.017), but not for OS. *Conclusions*: This study demonstrated that the NPS measured at diagnosis may serve as a useful independent prognostic factor for cancer progression in patients with cervical cancer.

## 1. Introduction

Cervical cancer is the fourth most common cancer among women worldwide [1]. Statistics from the National Cancer Institute indicated that the diagnosis and mortality rates of cervical cancer had gradually decreased since 1992, starting with a diagnosis rate of 11.1 per 100,000 and a mortality rate of 3.5 per 100,000 [2]. Up to the year 2020, the incidence and mortality rates of cervical cancer exhibited a declining trend, reaching 6.4 and 2.2 per 100,000 population, respectively. However, in 2021, a reversal in this trend was observed, with the incidence and mortality rates increasing to 6.9 and 2.3 per 100,000, respectively [2]. The American Cancer Society’s statistical projections indicated that approximately 13,820 new cervical cancer cases were expected to be diagnosed in the United States in 2024, and that an estimated 4360 deaths would occur, reflecting the continued clinical and epidemiological significance of the disease [3]. Recent advancements in cervical cancer screening and the widespread implementation of human papillomavirus (HPV) vaccination have led to a positive outcome, with a reduction in cervical cancer incidence by more than half in the 2000s compared to the 1970s [3]. However, there is a critical need to identify more effective prognosis factors, given the cervical cancer recurrence rates of approximately 11–22% in stages IB to IIA and 28–70% in stages IIB to IV [4]. This will enable the selection of patients at a high risk for recurrence and facilitate the development of optimal treatment plans to improve clinical outcomes.

Currently, squamous cell carcinoma antigen (SCC-Ag), carbohydrate antigen 19-9 (CA 19-9), and carcinoembryonic antigen (CEA) are commonly used to assess treatment response in patients with cervical cancer. SCC-Ag is a quantitative serum marker elevated in squamous cell carcinomas of the cervix, lung, and esophagus [5]. CA 19-9, produced in the pancreatic and biliary epithelium as well as in the gastrointestinal and uterine mucosa, is elevated in various malignancies including pancreatic, biliary, cervical, and ovarian cancers [6]. CEA, synthesized by gastrointestinal epithelial cells, is elevated in breast, lung, gastric, pancreatic, and ovarian cancers [7]. However, the clinical utility of these markers in cervical cancer is limited due to their low specificity. Their levels may also be elevated in non-malignant conditions such as pelvic inflammatory disease, peptic ulcers, liver and renal diseases, and thyroid dysfunction [5,6,7], thereby limiting their prognostic value in cervical management.

Inflammatory responses in the cancer environment can induce angiogenesis of cancer cells and immunosuppression in the body, which can provide favorable conditions for cancer proliferation and metastasis [8]. Inflammatory scoring systems such as the systemic inflammation score (SIS) and neutrophil-to-lymphocyte ratio (NLR) have been developed to quantify the degree of inflammation. Several studies have reported a significant positive relationship between these scores and the prognosis of colorectal, gastrointestinal, hepatocellular, and esophageal cancers [9,10,11,12]. Additionally, the nutritional and immune status of cancer patients is also an important factor affecting prognosis, and several have shown that poor nutritional and immune status is associated with poor prognosis using nutritional status scoring systems such as prognostic nutritional index (PNI) and controlling nutritional status (CONUT) [13,14].

In line with this trend, the Naples prognostic score (NPS) was developed as a scoring system that integrates serum albumin, total cholesterol, NLR, and lymphocyte-to-monocyte ratio (LMR) to comprehensively assess a patient’s nutritional, immune, and inflammatory status [15]. For these reasons, our research was focused on investigating the potential of the NPS to serve as a valuable prognostic indicator among the various known prognostic factors in cervical cancer. Since the first article by Galizia et al. in 2017, which demonstrated the value of the NPS as a prognostic factor by reporting a significant positive relationship between the NPS and colorectal cancer aggravation [16], several articles have been published reporting that the NPS is a significant prognostic factor not only in colorectal cancer but also in other various malignancies such as gallbladder, hepatocellular, gastric cancer, and pancreatic cancers [17,18,19,20]. Nonetheless, only few studies have evaluated the significance of the NPS as a prognostic factor in patients with cervical cancer. Therefore, this study aimed to evaluate the association between the NPS measured at the time of diagnosis and survival outcomes and to assess its prognostic significance in cervical cancer.

## 2. Materials and Methods

### 2.1. Patient Selection and Data Collection

Data of 178 patients diagnosed with cervical cancer at Korea University Anam Hospital from January 2019 to December 2023 were retrospectively evaluated. Patients were diagnosed with cervical cancer, through a cytological test such as a cervical Pap smear and histological methods such as colposcopy and punch biopsy. Liquid-based cytology or conventional Pap smears were used for initial screening, followed by colposcopic examination in patients with abnormal cytologic findings. Colposcopy-directed punch biopsy was performed in all cases to confirm the diagnosis, and the final histological diagnosis of either squamous cell carcinoma or adenocarcinoma of cervical cancer was made based on biopsy results. To select 178 participants from the 225 patients diagnosed with cervical cancer, patients who had incomplete clinical data, were lost to postoperative follow-up, and had other concurrent malignancies, inflammatory disease, or other suspected infections were excluded. A flowchart of study participant selection is presented in Figure 1.

Patients initially diagnosed with cervical cancer were hospitalized and underwent routine blood and evaluative tests, including abdominal computed tomography (CT), chest CT, pelvic magnetic resonance imaging (MRI), gastrofiberscopy, total colonofiberscopy, mammography, and positron emission tomography-CT (PET-CT). Among the various aforementioned evaluation tests, routine blood tests included a complete blood count with differential and blood chemistry examinations, as well as tumor marker tests for SCC-Ag, CA 19-9 and CEA. For NPS calculation, the serum albumin, total cholesterol, neutrophil, lymphocyte, and monocyte levels were determined using the aforementioned tests. Subsequently, these values were scored as follows: (i) score of 0: serum albumin ≥ 4 mg/dL, total cholesterol > 180 mg/dL, NLR ≤ 2.96, and LMR > 4.44 and (ii) score of 1: serum albumin < 4 mg/dL, total cholesterol ≤ 180 mg/dL, NLR > 2.96, and LMR ≤ 4.44 [16]. The NPS was defined as the sum of the aforementioned scores, resulting in a total NPS of 0, 1, 2, 3 or 4 [16]. The optimal cut-off value for the NPS was determined through receiver operating characteristic (ROC) curve analysis of disease-free survival (DFS). Based on this, patients were categorized into a low-NPS group (NPS 0 or 1) and a high-NPS group (NPS ≥ 2).

Treatment methods for patients with cervical cancer varied, depending on the stage and fertility preservation. The 2018 International Federation of Gynecology and Obstetrics (FIGO) staging system for cervical cancer is presented in Table A1 [21]. The 2018 FIGO staging system was used to classify cervical cancer in all patients. Although the FIGO system was updated in 2023, the 2018 version was retained to ensure consistency in staging, as all patients were initially diagnosed and treated based on this classification. Applying the 2023 staging retrospectively was not feasible due to the lack of uniform imaging and pathological data required for accurate reclassification. After determining the FIGO stage of cervical cancer in each participant, whose with stage I disease (including IA and IB) were classified as the early-stage group, while those with stage II or higher (including IIA and IIB) were categorized as the advanced-stage group. Treatment was based on FIGO stage and is summarized in Figure 2. Patients with stage IA2 or IB1 who wished to preserve fertility underwent radical trachelectomy. Those with stage IA2 who did not desire fertility preservation received modified radical hysterectomy, while radical hysterectomy was performed for stage IB1 or higher. Concurrent chemoradiotherapy (CCRT) was considered for stage IB3-IIB, and chemotherapy was used for stage IIIC2 or higher with para-aortic lymph node involvement [21]. Patients not requiring adjuvant therapy were followed every 3 months during the first year, every 6 months for the next 2 years, and annually thereafter, with abdominal/chest CT and tumor marker (SCC, CA 19-9, CEA). CCRT included radiotherapy five day per week and weekly cisplatin (40 mg/m^2^), with response assessed by pelvic MRI after 23–28 sessions. Chemotherapy commonly involved paclitaxel with cisplatin or carboplatin, with or without bevacizumab. In advanced or recurrent cases unsuitable for surgery or radiotherapy, immune checkpoint inhibitors such as pembrolizumab or nivolumab were also used. Chemotherapy was administered every 3 weeks, and treatment response was evaluated after three cycles. If progression was observed, alternative regimens were considered. For survival analysis, DFS was calculated as the time from the date of cervical cancer diagnosis to the date of recurrence, as confirmed by imaging. Overall survival (OS) was defined as the time from the date of cervical cancer diagnosis to the last follow-up or death.

### 2.2. Statistical Analysis

Continuous variables were analyzed and compared between the low- and high-NPS groups using Student’s *t*-test, and categorical variables were examined using the chi-square test or Fisher’s exact test. DFS and OS were analyzed using the Kaplan–Meier method, and time-to-event outcomes were compared using the log-rank test. Cox proportional hazards models were used for both univariate and multivariate analyses to identify factors associated with cervical cancer recurrence and overall survival (OS). Variables that showed statistical significance in the univariate analysis and were considered clinically relevant were included in the multivariate analysis for adjustment. The final multivariate model was constructed using a backward stepwise selection method. All statistical analyses were performed using SPSS Statistics for Windows version 25.0 (SPSS Inc., Chicago, IL, USA), with statistical significance set at *p* < 0.05. In addition, ROC curve analysis to select the optimal cut-off value of NPS was performed using MedCalc Statistical Software version 20.218 (MedCalc Software Ltd., Ostend, Belgium), and the Youden Index was used to identify the threshold that maximized the sum of sensitivity and specificity.

## 3. Results

The ROC analysis identified NPS ≥ 2 as the optimal cutoff value with the highest Youden Index. The area under the curve (AUC) was 0.69, with a sensitivity of 66.1% and specificity of 64.8%. Based on these results, among the 178 patients included in the study, 98 were categorized into the low-NPS group and 80 into the high-NPS group. The results of the ROC curve for predicting DFS according to NPS status are presented in Figure 3, and the characteristics of the participants are summarized in Table 1. The mean patient age was 55.54 ± 14.88 years (range, 22–98 years). The average age was 53.05 ± 14.09 years in the low-NPS group and 58.60 ± 13.35 years in the high-NPS group, with the high-NPS group being significantly older than the low-NPS group. The mean height and weight were significantly higher in the low-NPS group than in the high-NPS group. Furthermore, the rates of menopausal status, comorbidities, squamous cell type, advanced-stage group, tumor size of 4cm or more, pelvic or para-aortic lymph node (LN) involvement, positive HPV infection, and adjuvant chemotherapy status were significantly higher in the high-NPS group than in the low-NPS group. According to our results, the positive rate of HPV infection was 44.3% in the low NPS group and 61.3% in the high NPS group, which is lower than the generally reported positive rate of HPV infection in cervical cancer patients of 90–98%. Although HPV testing was performed at Korea University Anam hospital using cervical specimens collected by real-time polymerase chain reaction (PCR) method, which ensured consistency in terms of instrumentation, it cannot be ruled out that various factors such as the retrospective nature of our study and the possibility of false negatives due to specimen preservation conditions may have affected the results. Among the tumor markers (namely, SCC-Ag, CA 19-9 and CEA), the SCC-Ag and CEA levels did not significantly differ between the two groups. However, the proportion of patients with abnormally elevated CA 19-9 levels was significantly higher in the high-NPS group than in the low-NPS group. Among the study population, pembrolizumab, an immune checkpoint inhibitor, was administered to only nine patients (5.1%)—one patient in the low NPS group and eight in the high NPS group. Bevacizumab was administered to 33 patients (19.5%), including 13 in the low NPS group and 20 in the high NPS group. Due to the relatively low number of cases, conducting subgroup or stratified analyses based on these treatments was deemed statistically unreliable and potentially prone to misinterpretation. Furthermore, since the primary objective of this study was to evaluate prognostic indicators at the time of diagnosis rather than treatment response, the use of agents such as immune checkpoint inhibitor or bevacizumab—typically administered in cases of advanced or progressive disease—was considered to have limited relevance to initial prognostic assessment. Therefore, these treatments were not included as covariates in the statistical analysis.

Kaplan–Meier curve analysis was conducted to compare DFS and OS based on NPS status, as shown in Figure 4. The median follow-up time was 30.24 ± 16.71 months. During the follow-up period, the recurrence and mortality rates were 31.5% and 9.0% in the low- and high-NPS groups, respectively. The 2-year DFS rates were 86.2% in the low-NPS group and 74.0% in the high-NPS group. The 2-year OS rates were 97.8% in the low-NPS group and 88.0% in the high-NPS group. A comparison of survival plots between the two groups revealed that the low-NPS group had significantly prolonged DFS and OS compared with the high-NPS group (DFS, *p* < 0.001; OS, *p* = 0.02).

To explore the risk factors for cervical cancer recurrence, univariate and multivariate analyses were performed using the Cox proportional hazards model (Table 2). The univariate Cox proportional analysis of DFS showed that comorbidities, FIGO stage, tumor size, LN metastasis status, HPV infection status, adjuvant chemotherapy status, number of chemotherapies, radical hysterectomy status, CA 19-9 levels, and NPS status were significantly positively associated with cervical cancer recurrence. Multivariate Cox proportional analysis was performed with adjustment for age, comorbidities, FIGO stage, tumor size, HPV infection status, and radical hysterectomy status, among the factors that were statistically significant in the univariate Cox analysis. The results indicated that the high-NPS group (adjusted hazard ratio [HR], 1.98; 95% confidence interval [CI], 1.131, 3.455; *p* = 0.017), advanced-FIGO stage group (adjusted HR, 5.84; 95% CI, 2.483, 13.723; *p* < 0.001), pelvic LN metastasis (adjusted HR, 2.32; 95% CI, 1.157, 4.633; *p* = 0.018), pelvic LN and para-aortic LN metastases (adjusted HR, 3.72; 95% CI, 1.725, 8.033; *p* = 0.001), adjuvant chemotherapy status (adjusted HR, 15.42; 95% CI, 7.426, 32.020; *p* < 0.001), and increased number of chemotherapies (adjusted HR, 1.09; 95% CI, 1.059, 1.115; *p* < 0.001) were statistically significant risk factors associated with cervical cancer recurrence. In addition, univariate and multivariate analyses using the Cox proportional hazards model were performed to identify risk factors influencing OS in cervical cancer. However, no statistically significant risk factors correlating with the OS of patients with cervical cancer were found. The detailed statistical results are provided in the Table A2. To evaluate potential multicollinearity among the covariates included in the multivariate Cox proportional hazards model, variance inflation factors (VIFs) were calculated. All variables demonstrated VIF values below 2.0, indicating that multicollinearity was not a significant concern in the model. The results of the VIFs analysis are presented in Table A3.

## 4. Discussion

This study confirmed that a high NPS (≥2) measured at the time of cervical cancer diagnosis was a significant risk factor for recurrence. However, no significant association between the OS of patients with cervical cancer and high NPS was identified. In the multivariate Cox proportional hazards analysis, FIGO stage was included among the adjusted variables. Even after adjusting for key clinical factors such as age, comorbidities, FIGO stage, tumor size, HPV infection status, and radical hysterectomy status, NPS remained a significant independent prognostic factor for DFS in cervical cancer. This finding suggests that NPS may provide independent prognostic value beyond traditional indicators of tumor burden in cervical cancer. Furthermore, although NPS showed a significant association with DFS, it did not demonstrate a similar association with OS. This discrepancy may be due to limited follow up duration or the influence of post-recurrence treatments on OS. It is also possible that NPS is more closely related to early recurrence than to long term survival. Further research is needed to clarify whether biological differences between DFS and OS underlie these findings. To the best of our knowledge, this is one of the first studies to demonstrate the association between NPS and DFS in cervical cancer, as measured by laboratory results at the time of cervical cancer diagnosis.

Several studies have reported the close relationship between the immune system and cancer progression [18,22]. Neutrophils, key effectors of inflammation, contribute to cancer progression by secreting serine proteases and metalloproteinases (MMP), which degrade extracellular matrix (ECM) proteins and facilitate tumor migration and invasion [22]. Alveolar neutrophils release hepatocyte growth factors and promote lung cancer cell migration and progression [22]. Additionally, neutrophil-derived proteins induce epithelial–mesenchymal transition (EMT), weaken cell junctions, and enhance tumor spread [22]. Neutrophils also secrete transforming growth factor (TGF)-beta, upregulating transcription factors such as Snail1/2, Zeb1/2, and Twist1, which drive EMT and promote lung adenocarcinoma development [23]. Lymphocytes, including T, B, and natural killer (NK) cells, play a key role in immune activation by distinguishing foreign from self-antigen [18]. In the tumor microenvironment, B cells produce tumor-reactive antibiotics, enhance NK cell activity, and promote macrophage phagocytosis, thereby suppressing tumor development [18,24]. Studies have shown that high levels of pre-tumor and tumor-infiltrating B cells in cervical and lung cancers are associated with lower recurrence rates and improved survival [25,26]. However, B cells can also promote tumor growth. Tumor-secreted leukotriene B4 activates peroxisome proliferator-activated receptor (PPAR)-alpha in B cells, inducing their differentiation into regulatory B cells. These cells produce TFG-beta, which converts naïve cluster of differentiation (CD)4+ T cells and forkhead box P3 (FOXP3)+ regulatory T cells, suppresses NK and CD8+ T cell activity, and ultimately facilitates tumor growth [24]. Thus, lymphocytes play dual roles in cancer, contributing to both tumor suppression and progression. Monocytes differentiate into tumor-associated macrophages (TAMs), which promote tumor progression by secreting vascular endothelial growth factor (VEGF), platelet-derived growth factor (PDGF), chemokines, proteases, and MMPs for angiogenesis, ECM remodeling, and growth factor release [27]. However, immune cells such as neutrophils, lymphocytes, and monocytes are influenced by host conditions, limiting the predictive power of individual markers [18]. To address this, inflammation-related prognostic scores, including the NLR and LMR, have gained attention. A meta-analysis by Zou found that an NLR cutoff above 2.46 correlated with poorer DFS and OS in cervical cancer, independent of lesion extent or primary treatment [28]. Other retrospective studies have similarly reported a high NLR for poor prognosis [29,30,31]. Conversely, high LMR have been associated with longer DFS and OS, suggesting their potential as favorable prognostic markers for cervical cancer [32,33].

Malnourished patients with cancer exhibit reduced tolerance to treatment, which negatively affects their outcomes [17]. Serum albumin and cholesterol levels are key nutritional indicators, and studies have related poor nutrition to poor cancer prognosis [14,34]. Low albumin levels indicate nutritional deficiency and systemic inflammation, which impair immune cell receptor mobility and signal transduction, thereby weakening the antitumor response and potentially promoting cancer progression [18]. Retrospective studies on patients with stage I cell lung cancer or endometrial cancer have associated hypoalbuminemia with decreased DFS and OS, suggesting its role as a negative prognostic factor [35,36]. Cholesterol influences cancer progression. Low cholesterol levels increase interleukin (IL)-6, which binds to membrane receptors and gp-130 proteins and activates Janus kinase (JAK), triggering signal transducer and activator of transcription 3 (STAT3) signaling. This inhibits mitochondrial apoptosis in tumor cells, whereas JAK-mediated activation of mitogen-activated protein kinase (MAPK) induces hypoxia and promotes inflammation, angiogenesis, tumor proliferation, and migration [37,38]. A case–control study in China related low total cholesterol levels to an increased risk of gastric cancer [39]. Similarly, a Korean prospective study found an inverse relationship between cholesterol levels and overall cancer incidence [40]. A United Kingdom (UK) Biobank study also associated low cholesterol levels with an increased risk of plasma cell neoplasm [41].

Immune cells, including neutrophils, lymphocytes, and monocytes, along with nutritional markers, such as albumin and cholesterol, are useful for predicting cancer prognosis. However, relying solely on single markers is insufficient for accurate prognostic assessment. The NPS, which integrates serum albumin level, total cholesterol level, NLR, and LMR, is an effective prognostic tool. Our findings support this, showing that cervical cancer patients with NPS ≥ 2 at diagnosis had a significantly increased risk of recurrence. Several studies have linked a higher NPS to poor cancer survival across various malignancies. Retrospective studies identified elevated NPS as a significant risk factor for OS in gallbladder (HR, 3.417; *p* < 0.05), gastric (HR, 3.45; *p* < 0.001), and pancreatic (HR, 1.884; *p* < 0.001) cancers [17,19,20]. Similarly, Xie et al. demonstrated that a high NPS was a significant risk factor for DFS (HR, 2.002; *p* = 0.003) and OS (HR, 2.608; *p* = 0.004) in hepatocellular carcinoma [18]. A study of patients with endometrial cancer undergoing surgery also associated high NPS with poor DFS (HR, 6.725; *p* = 0.011) and OS (HR, 4.066; *p* = 0.039) [15]. Zhang et al. confirmed that a higher NPS was significantly associated with poor DFS (HR, 2.738; *p* < 0.001) and OS (HR, 2.799; *p* < 0.001) in patients with cervical cancer undergoing CCRT [42]. While this study focused exclusively on locally advanced cervical cancer, our study included all patients with cervical cancer, regardless of the stage. Additionally, Zhang et al. assessed the systemic inflammation index (SII) and PNI as prognostic factors, demonstrating that both SII (DFS: HR, 1.134; *p* = 0.045) (OS: HR, 1.298; *p* = 0.044) and PNI (DFS: HR, 1.457; *p* = 0.022) (OS: HR, 1.611; *p* = 0.019) were also associated with DFS and OS [42]. Among the previously discussed studies, Xie et al. and Li et al. evaluated the prognostic value of the NPS alongside the SIS, PNI, and CONUT. However, SIS, PNI, and CONUT scores were not found to be statistically significant independent predictors of survival outcomes [15,18]. The SIS, calculated using serum albumin and LMR, has been primarily studied in gastric cancer, where a high SIS is associated with poor survival. It has also been identified as a risk factor in cholangiocarcinoma, prostate cancer and esophageal cancer [43,44,45,46,47,48]. The PNI, calculated as 10 × serum albumin (g/dL) + (0.005 × total lymphocyte count/mm^3^), is associated with increased recurrence risk and shorter survival in breast, prostate, and endometrial cancers [49,50,51,52]. The CONUT score, which is used to assess malnutrition risk, has been associated with higher mortality in the elderly and long-term hospitalized patients. It is determined by summing the scores assigned to albumin, cholesterol, and lymphocyte levels [53,54]. The SII, defined as platelet × NLR, has been correlated with increased recurrence and shorter survival in hepatocellular carcinoma and gastric cancer [55,56]. Although various prognostic markers exist, NPS provides a more comprehensive assessment of immune and nutritional status, and several studies have confirmed that a high NPS is associated with poor survival. However, Pian et al. found no significant relationship between NPS and DFS or OS [57]. This suggests that the prognostic value of NPS remains inconsistent, and a definitive conclusion is yet to be established. To date, numerous studies have compared NPS and survival outcomes in various patients with cancer. In this context, It is also important to consider the impact of different cancer types’ cellular environment on NPS. A study conducted by Alvarez et al. reported that in the HPV-infected cervical cancer microenvironment, the levels of T helper (Th)2-type cytokines, such as IL-10, are elevated, leading to increased neutrophil expression in response to specific antigens. Conversely, the expression of interferon (IFN)-gamma, which facilitates lymphocyte activation, is reduced, resulting in decreased lymphocyte levels [58]. These findings suggest that HPV-infected cervical cancers may create conditions that promote an increased NLR, thereby contributing to a higher NPS. This further supports the prognostic significance of NPS, even when considering the cellular environment of cervical cancer.

This study has some limitations. First, as a retrospective study, it did not allow longitudinal follow-up of the participants. Second, a selection bias might have been introduced during participant selection. Third, our study was conducted at a single institution; consequently, the number of participants was smaller than that included in multicenter studies. In light of this limitation, future multicenter studies should be planned to validate and extend the findings of the present study. Fourth, the primary aim of this study was to evaluate the prognostic value of the NPS assessed at the time of cervical cancer diagnosis. Therefore, treatments such as immune checkpoint inhibitors and bevacizumab—typically administered in advanced or recurrent disease—were not included as covariates in the statistical analysis, as their relevance to initial prognostic assessment was considered limited. However, the possibility that these therapies may have influenced survival outcomes cannot be completely excluded and, thus, potential confounding effects may exist. For these reasons, the statistical results of our study may not be sufficient to generalize the NPS as a prognostic factor for predicting patient outcomes in cervical cancer. Nevertheless, our study has strength in that it is one of the few studies to evaluate the association between the NPS and survival outcomes and to assess the prognostic value of NPS at the time of cervical cancer diagnosis. In order to generalize the causal relationship between NPS and recurrence or overall survival in cervical cancer patients and the value of NPS as a prognostic factor, longitudinal follow up studies are needed. Furthermore, given its simplicity and cost-effectiveness, NPS could serve as a practical tool to help identify high-risk patients at diagnosis who may benefit from closer surveillance or more personalized follow-up strategies in clinical practice.

## 5. Conclusions

The results of this study confirmed that a high NPS is a significant independent risk factor for poor DFS in patients with cervical cancer. These findings suggest that the NPS, which reflects both inflammatory and nutritional status, may serve as a valuable prognostic biomarker for identifying patients at higher risk of recurrence. Incorporating the NPS into clinical practice could help stratify patients more effectively and guide individualized follow-up and therapeutic strategies.

## Figures and Tables

**Figure 1 medicina-61-01381-f001:**
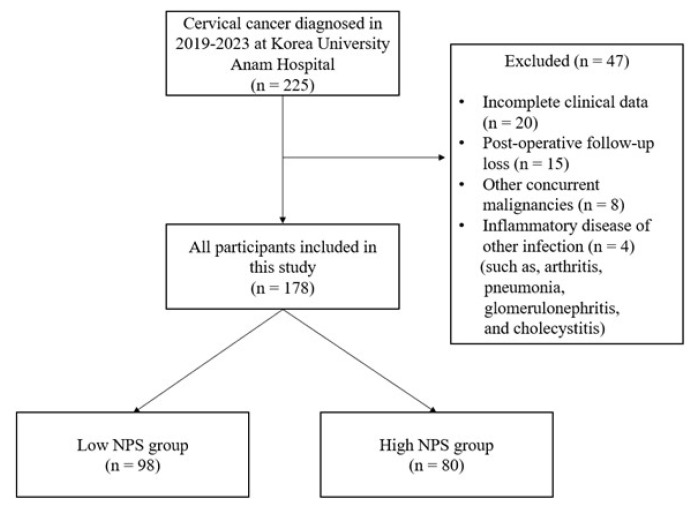
Flowchart of study participants selection. NPS, Naples prognostic score.

**Figure 2 medicina-61-01381-f002:**
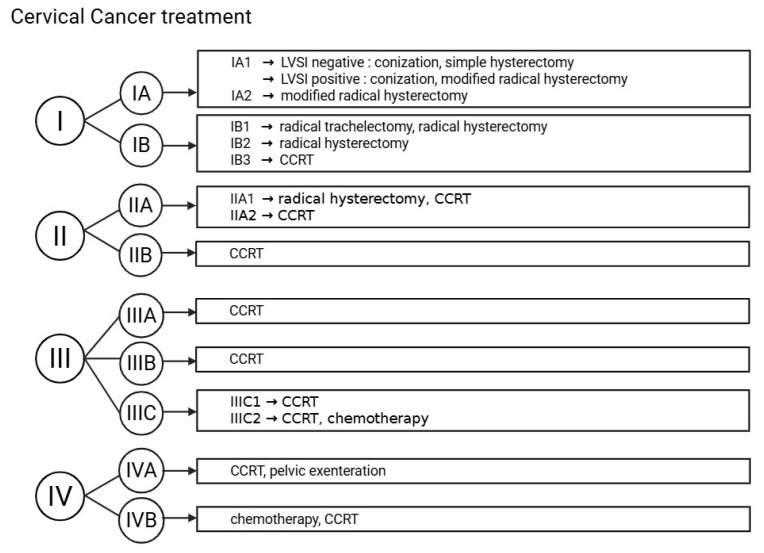
Cervical cancer treatment algorithm according to FIGO stage. LVSI, lympho-vascular space invasion; CCRT, concurrent chemoradiotherapy; FIGO, Federation of Gynecology and Obstetric.

**Figure 3 medicina-61-01381-f003:**
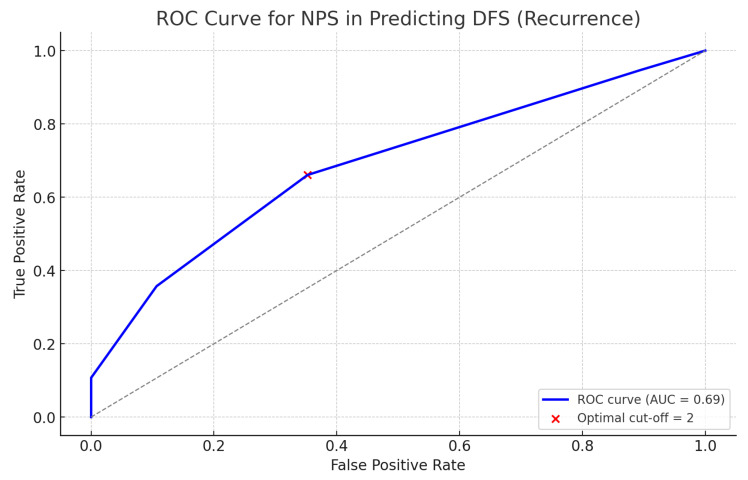
ROC curve for predicting DFS according to NPS status. The optimal cut-off value (NPS, 2) was determined using the Youden Index (sensitivity, 66.1%; specificity, 64.8%), with an AUC of 0.69). ROC, Receiver Operating Characteristic; DFS, disease-free survival; NPS, Naples prognostic score; AUC, Area Under the Curve.

**Figure 4 medicina-61-01381-f004:**
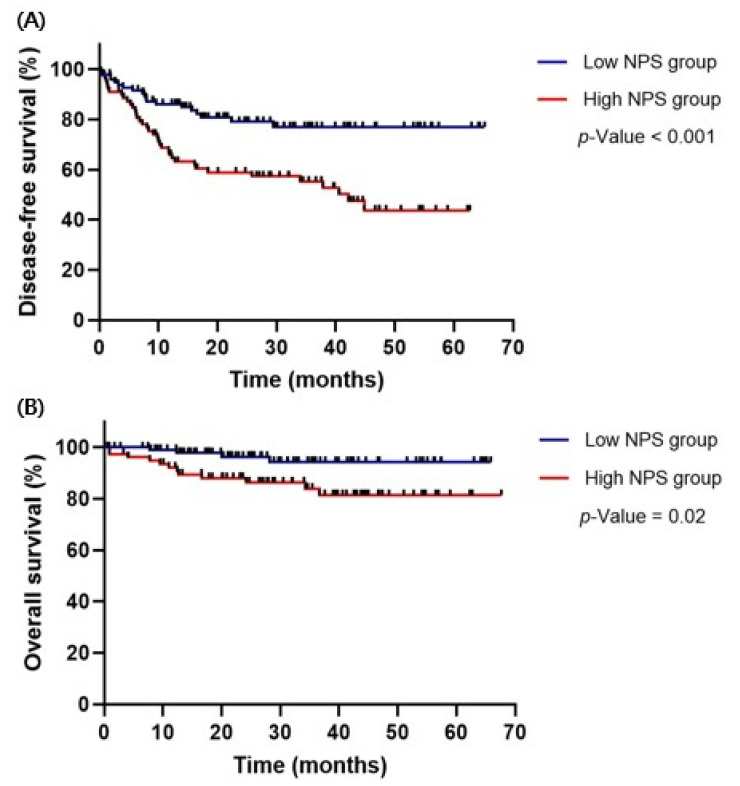
Survival plot by NPS status according to the Cox proportional hazards model. (**A**) DFS plot and (**B**) OS plot. Survival outcomes were stratified by NPS group (low versus high), with higher NPS showing a trend toward worse prognosis. DFS, disease-free survival; OS, overall survival.

**Table 1 medicina-61-01381-t001:** Characteristics of the study participants according to the NPS status of cervical cancer.

	Low-NPS Group (N = 98)	High-NPS Group (N = 80)	*p*-Value
Age (years)	53.05 ± 14.09	58.60 ± 13.35	0.013
Height (cm)	158.52 ± 5.89	154.33 ± 7.16	<0.001
Weight (kg)	60.44 ± 8.84	56.53 ± 11.91	0.016
BMI (kg/m^2^)	24.33 ± 4.45	23.66 ± 4.30	0.311
Parity			0.080
Nulliparous	26 (26.5%)	12 (15.0%)	
Multiparous	72 (73.5%)	68 (85.0%)	
Menopausal status			0.026
No	40 (40.8%)	20 (25.0%)	
Yes	58 (59.2%)	60 (75.0%)	
Comorbidities			0.016
No	72 (73.5%)	45 (56.3%)	
Yes	26 (26.5%)	35 (43.7%)	
Histology			0.010
Squamous	69 (71.1%)	63 (78.8%)	
Adenocarcinoma	28 (28.9%)	13 (16.2%)	
Others	0 (0.0%)	4 (5.0%)	
FIGO stage			0.002
Early-stage group	48 (49.0%)	21 (26.3%)	
Advanced-stage group	50 (51.0%)	59 (73.7%)	
Tumor size			0.031
≤4 cm	60 (61.2%)	36 (45.0%)	
>4 cm	38 (38.8%)	44 (55.0%)	
LN metastasis status			0.002
Negative	70 (71.4%)	39 (48.8%)	
PLND metastasis	20 (20.4%)	21 (26.2%)	
PLND and PALND	8 (8.2%)	20 (25.0%)	
metastases			
HPV infection status			0.025
No	54 (55.7%)	31 (38.7%)	
Yes	43 (44.3%)	49 (61.3%)	
CCRT status			0.511
No	39 (39.8%)	28 (35.0%)	
Yes	59 (60.2%)	52 (65.0%)	
Adjuvant chemotherapy status			0.001
No	81 (82.7%)	48 (60.0%)	
Yes	17 (17.3%)	32 (40.0%)	
Number of chemotherapies	1.09 ± 2.59	4.36 ± 7.71	<0.001
Radical hysterectomy status			0.222
No	60 (61.2%)	56 (70.0%)	
Yes	38 (38.8%)	24 (30.0%)	
Trachelectomy status			0.381
No	94 (95.9%)	79 (98.8%)	
Yes	4 (4.1%)	1 (1.2%)	
SCC-Ag level			0.334
≤2.0	68 (69.4%)	50 (62.5%)	
>2.0	30 (30.6%)	30 (37.5%)	
CA 19-9 level			0.016
≤37	87 (88.8%)	60 (75.0%)	
>37	11 (11.2%)	20 (25.0%)	
CEA level			0.232
≤4.6	88 (89.8%)	67 (83.8%)	
>4.6	10 (10.2%)	13 (16.3%)	

Note: Values are presented as mean ± standard deviation or N (%). BMI, body mass index; FIGO, International Federation of Gynecology and Obstetrics; LN, lymph node; PLND, pelvic lymph node; PALND, para-aortic lymph node; HPV, human papillomavirus; CCRT, concurrent chemoradiotherapy; SCC-Ag, squamous cell carcinoma antigen; CA, carbohydrate antigen; CEA, carcinoembryonic antigen.

**Table 2 medicina-61-01381-t002:** Cox proportional hazards model of disease-free survival in the univariate and multivariate analyses.

Risk Factors for DFS						
	HR	95% CI	*p*-Value	HR *	95% CI	*p*-Value
Age (years)	1.02	(0.999, 1.036)	0.058	0.99	(0.969, 1.012)	0.375
Height (cm)	0.97	(0.927, 1.004)	0.078	1.01	(0.954, 1.050)	0.960
Weight (kg)	0.99	(0.963, 1.015)	0.409	0.99	(0.973, 1.027)	0.985
BMI	0.99	(0.929, 1.049)	0.671	0.99	(0.926, 1.065)	0.852
Parity						
Nulliparous	1.00	Reference		1.00	Reference	
Multiparous	2.21	(1.001, 4.880)	0.058	1.66	(0.670, 4.087)	0.275
Menopausal status						
No	1.00	Reference		1.00	Reference	
Yes	1.60	(0.886, 2.897)	0.119	0.96	(0.401, 2.298)	0.926
Comorbidities						
No	1.00	Reference		1.00	Reference	
Yes	1.91	(1.130, 3.237)	0.016	1.63	(0.919, 2.893)	0.094
Histology						
Squamous	1.00	Reference		1.00	Reference	
Adenocarcinoma	1.96	(1.112, 3.468)	0.020	2.56	(1.442, 4.550)	0.001
Others	2.40	(0.575, 10.001)	0.230	1.78	(0.423, 7.524)	0.430
FIGO stage						
Early-stage group	1.00	Reference		1.00	Reference	
Advanced-stage group	6.79	(2.909, 15.855)	<0.001	5.84	(2.483, 13.723)	<0.001
Tumor size						
≤4 cm	1.00	Reference		1.00	Reference	
>4 cm	2.08	(1.213, 3.551)	0.008	0.87	(0.462, 1.642)	0.669
LN metastasis status						
Negative	1.00	Reference		1.00	Reference	
PLND metastasis	3.74	(2.010, 6.975)	<0.001	2.32	(1.157, 4.633)	0.018
PLDN and PALND	5.54	(2.833, 10.848)	<0.001	3.72	(1.725, 8.033)	0.001
metastases						
HPV infection status						
No	1.00	Reference		1.00	Reference	
Yes	2.02	(1.161, 3.511)	0.013	1.53	(0.844, 2.764)	0.162
CCRT status						
No	1.00	Reference		1.00	Reference	
Yes	1.69	(0.921, 3.092)	0.090	0.54	(0.266, 1.076)	0.079
Adjuvant chemotherapy status						
No	1.00	Reference		1.00	Reference	
Yes	20.55	(10.472, 40.322)	<0.001	15.42	(7.426, 32.020)	<0.001
Number of chemotherapies	1.10	(1.078, 1.131)	<0.001	1.09	(1.059, 1.115)	<0.001
Radical hysterectomy status						
No	1.00	Reference		1.00	Reference	
Yes	0.54	(0.295, 0.991)	0.047	1.43	(0.673, 3.015)	0.355
Trachelectomy status						
No	1.00	Reference		1.00	Reference	
Yes	0.63	(0.087, 4.583)	0.651	2.69	(0.309, 23.456)	0.370
SCC-Ag level						
≤2.0	1.00	Reference		1.00	Reference	
>2.0	1.43	(0.833, 2.463)	0.194	0.80	(0.428, 1.493)	0.483
CA 19-9 level						
≤37	1.00	Reference		1.00	Reference	
>37	2.46	(1.378, 4.405)	0.002	1.55	(0.851, 2.834)	0.151
CEA level						
≤4.6	1.00	Reference		1.00	Reference	
>4.6	1.54	(0.751, 3.138)	0.240	0.96	(0.458, 2.010)	0.913
NPS status						
Low NPS	1.00	Reference		1.00	Reference	
High NPS	1.89	(1.422, 2.465)	<0.001	1.98	(1.131, 3.455)	0.017

Note: HR, hazard ratio; HR *, hazard ratio adjusted for age, comorbidities, FIGO stage, tumor size, HPV infection status, and radical hysterectomy status; CI, confidence interval; DFS, disease-free survival; BMI, body mass index; FIGO, International Federation of Gynecology and Obstetrics; LN, lymph node; PLND, pelvic lymph node; PALND, para-aortic lymph node; HPV, human papillomavirus; CCRT, concurrent chemoradiotherapy; SCC-Ag, squamous cell carcinoma antigen; CA, carbohydrate antigen; CEA, carcinoembryonic antigen; NPS, Naples prognostic score.

## Data Availability

The data and analysis code presented in this study are available on reasonable request from the corresponding author.

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
