# Peer review of "The Value of the Naples Prognostic Score at Diagnosis as a Predictor of Cervical Cancer Progression"

_medicina, 2025, doi:10.3390/medicina61081381_

Round 1

Reviewer 1 Report

Comments and Suggestions for Authors

I would like to congratulate the authors of their work, but I have a few concerns.

  1. Authors write at the beginning that the number of new cases and deaths due to cc has been falling since 1992 (11.1 and 3.5 per 100,000) and then that since 2021 there has been an increase in frequency (6.9 and 2.3 per 100,000). In that case, to what values ​​did these indicators fall before 2021? And if the decline since 1992 was more rapid and reached a plateau in 2021, please rephrase the sentence.
  2. “ […] approximately 4,360 patients area were expected to die from cervical cancer, which is not a negligible number” – please avoid such subjective terms, this is not scientific language
  3. In the introduction, please focus more on the methods of cervical cancer diagnostics, with a description of highly oncogenic HPV viruses associated with HPV-dependent diseases, such as cc. Please look at works comparing the sensitivity and specificity of advanced molecular methods

In addition, I would like to devote a few sentences to vaccinations, which are a benefit in prevention and are increasingly used to induce remission in patients already infected with HPV. 

Rokita W, Kedzia W, Pruski D, Friebe Z, Nowak-Markwitz E, Spaczyński R, Karowicz-Bilińska A, Spaczyński M. Comparison of the effectiveness of cytodiagnostics, molecular identification of HPV HR and CINtecPLUS test to identify LG SIL and HG SIL. Ginekol Pol. 2012 Dec;83(12):894-8. PMID: 23488290.

Instead, the authors devote an entire paragraph to Ca 19 9 or CEA markers, in the case of other cancers, whose sensitivity and specificity is very limited.

4.     Please clarify the methodology. "Patients were diagnosed with cervical cancer, including SCC or adenocarcinoma, through a cytological test such as a cervical Pap smear and histological methods such as colposcopy and punch biopsy." Was the cytology LBC or a smear taken on a slide? Was a cytological diagnosis sufficient to make a diagnosis of cervical cancer? Did each patient have a colposcopy and biopsy? There is too little information to assess the quality of the described methodology. This absolutely requires improvement.

5.     I propose that the FIGO classification of cervical cancer be included as a supplement and not as a table in the text, because it was not created by the authors.

6.     Instead of a long description in the text of which stage of cc requires which treatment, I propose to make a simplified diagram. It will be more readable and pleasant to read.

7.     Please check the tables carefully - for example - in the column "Low-NPS group" the data regarding HPV status and cervical cancer histology do not add up to 98 (100%) - if you do not have the data, please mark it!!

8.     How can you explain such a low percentage of cervical cancers unrelated to HPV infection? 44% where according to global data this is a correlation of 90-98% in the case of cervical cancer? Where was the HPV test taken - from blood or a smear from the cervical disc and canal?

9.     The results of the work may not be satisfactory, but the authors should be congratulated for their unconventional approach to the subject. Searching for new diagnostic methods and earlier detection of pre-cancerous changes is the second, after promoting vaccination, bastion against cervical cancer. 

After implementing the fixes, I recommend publishing.

Author Response

Manuscript;

The Value of the Naples Prognostic Score at Diagnosis as a Predictor of Cervical Cancer Progression (medicina-3719657)

Response to Reviewer 1 Comments

Dear reviewers

Thank you for giving us the opportunity to submit a revised draft of the manuscript “The Value of the Naples Prognostic Score at Diagnosis as a Predictor of Cervical Cancer Progression” for publication in the Medicina. We appreciate the time and effort that you and the reviewers dedicated to providing feedback on our manuscript and are grateful for the insightful comments on and valuable improvements to our paper.

We have incorporated most of the suggestions made by the reviewers. Any revisions to the manuscript be marked up using the track changes function at MS Word. In addition, the changed text color was changed to blue and displayed. Please see below, for a point-by-point response to the reviewers’ comments and concerns.

Point 1: Authors write at the beginning that the number of new cases and deaths due to cc has been falling since 1992 (11.1 and 3.5 per 100,000) and then that since 2021 there has been an increase in frequency (6.9 and 2.3 per 100,000). In that case, to what values ​​did these indicators fall before 2021? And if the decline since 1992 was more rapid and reached a plateau in 2021, please rephrase the sentence.

Point 2: approximately 4,360 patients area were expected to die from cervical cancer, which is not a negligible number” – please avoid such subjective terms, this is not scientific language

Response 2: Thanks for the good advice. As you pointed out, I agree that the phrase “which is not a negligible number” does not seem to be appropriate for thesis format. For this reason, I have reworded it to make it more appropriate for an English sentence rather than the original sentence.

Point 3: in the introduction, please focus more on the methods of cervical cancer diagnostics, with a description of highly oncogenic HPV viruses associated with HPV-dependent diseases, such as cc. Please look at works comparing the sensitivity and specificity of advanced molecular methods. In addition, I would like to devote a few sentences to vaccinations, which are a benefit in prevention and are increasingly used to induce remission in patients already infected with HPV. 

Rokita W, Kedzia W, Pruski D, Friebe Z, Nowak-Markwitz E, Spaczyński R, Karowicz-Bilińska A, Spaczyński M. Comparison of the effectiveness of cytodiagnostics, molecular identification of HPV HR and CINtecPLUS test to identify LG SIL and HG SIL. Ginekol Pol. 2012 Dec;83(12):894-8. PMID: 23488290.

Instead, the authors devote an entire paragraph to Ca 19 9 or CEA markers, in the case of other cancers, whose sensitivity and specificity is very limited.

Response 3: Thank you for your valuable suggestion. As you pointed out, the manuscript does mention that widespread implementation of cervical cancer screening and HPV vaccination has contributed to a decline in cervical cancer incidence. As this point is aleady addressed in the current manuscript, I respectfully believe that including the suggested reference may not be essential for the present contex. While the methods of cervical cancer diagnosis are described in the Methods section, I agree that more detailed explanataion would enhance the clarity of the study. Therefore, I have revised the Methods section to provide a more comprehensive description of the diagnostic procedure, in line with your recommendation. Also, I have streamlined the description of CA 19-9 and CEA in the Introduction. I have also revised the text to highlight the limited specificity of these tumor markers in the context cervical cancer prognosis.

Point 4: Please clarify the methodology. "Patients were diagnosed with cervical cancer, including SCC or adenocarcinoma, through a cytological test such as a cervical Pap smear and histological methods such as colposcopy and punch biopsy." Was the cytology LBC or a smear taken on a slide? Was a cytological diagnosis sufficient to make a diagnosis of cervical cancer? Did each patient have a colposcopy and biopsy? There is too little information to assess the quality of the described methodology. This absolutely requires improvement.

Response 4: Thank you for your good advice. As you mentioned, the process of diagnosing cervical cancer was only briefly presented in the existing manuscript, so I have revised the contents of the method section by describing it in more detail.

Point 5: I propose that the FIGO classification of cervical cancer be included as a supplement and not as a table in the text, because it was not created by the authors.

Response 5: Thank you for the good point. I agree that Table 1, which contains information about the FIGO stage of cervical cancer, would be more appropriately categorized as Table S1. For this reason, I have revised Table 1 to Table S1 and reflected in the manuscript.

Point 6: Instead of a long description in the text of which stage of cc requires which treatment, I propose to make a simplified diagram. It will be more readable and pleasant to read.

Response 6: Thank you for the good advice. As you mentioned, I have described the treatment of cervical cancer in the manuscript, but I have modified it to be presented as Figure 3 to make it easier to understand at a glance.

Point 7: Please check the tables carefully - for example - in the column "Low-NPS group" the data regarding HPV status and cervical cancer histology do not add up to 98 (100%) - if you do not have the data, please mark it!!

Response 7: Thank you for pointing this out. I sincerely apologize for the misstatement and have corrected it to the correct statistical figure as you pointed out. 

Point 8: How can you explain such a low percentage of cervical cancers unrelated to HPV infection? 44% where according to global data this is a correlation of 90-98% in the case of cervical cancer? Where was the HPV test taken - from blood or a smear from the cervical disc and canal?

Response 8: Thank you for the good advice. As you mentioned, the prevalence of HPV infection in our study was lower than the general prevalence of HPV infection in cervical cancer, which is 90-98%. Although HPV testing was performed at Korea University Anam hospital using cervical specimens collected by real-time PCR method, which is consistent in terms of instrumentation, I cannot exclude the possibility that various factors such as the retrospective nature of our study and the possibility of false negatives due to specimen preservation status may have affected the results. For these reases, it is possible that a lower infection rate was reported compared to the previously known HPV infection rate in cervical cancer, and I have revised the results by adding the above information to the Results section.

Point 9: The results of the work may not be satisfactory, but the authors should be congratulated for their unconventional approach to the subject. Searching for new diagnostic methods and earlier detection of pre-cancerous changes is the second, after promoting vaccination, bastion against cervical cancer. 

Response 9: Thank you very much for your kind encouragement and positive feedback on our study. I sincerely appreciate the valuale comments you have provided, and I have revised the manuscript accordingly to improve its overall quality. Once again, thank you for your thoughtful review and support. 

Reviewer 2 Report

Comments and Suggestions for Authors

Thank you for the opportunity to review this article. I found the topic both informative and clinically relevant—especially as someone still in the early years of medical school and learning about oncology and prognostic markers.

The manuscript is clearly written and follows a logical structure. It introduces a meaningful clinical question: can the Naples Prognostic Score (NPS), which is based on basic lab values, help predict outcomes in cervical cancer? This feels relevant, especially since it's simple to calculate and widely accessible.

That said, I’d suggest updating some of the references, particularly in the introduction. A few are more than 5 years old, and there are recent studies discussing squamous cell carcinoma (SCC) antigen that might be helpful. For example: Zhu H. Squamous Cell Carcinoma Antigen: Clinical Application and Research Status. Diagnostics (Basel). 2022 Apr 24;12(5):1065. doi: 10.3390/diagnostics12051065. PMID: 35626221; PMCID: PMC9139199. or Bashar H. Hamoud, Dragoș E. Georgescu, Amalia L. Călinoiu, Ileana A. Văcăroiu, Advances in Squamous Cell Carcinoma Antigen and Cervix Cancer Relationship., Romanian Journal of Military Medicine, New Series, Vol. CXXVI, No. 1/2023

Including this kind of recent reference could make the introduction more up to date and show ongoing interest in biomarker research.

The methods appear sound and easy to understand. The Naples Prognostic Score is explained clearly, and the grouping of patients into high and low NPS is logical. The use of survival analysis and regression models seems appropriate, although I admit some of the more advanced stats are beyond my current training level.

Still, the authors might consider briefly explaining why they chose NPS over similar prognostic scores like the PNI or CONUT, especially since all of them are derived from immune-nutritional status.

The methodology section is well-detailed, and I think someone could replicate this study in a similar clinical setting. It’s great that the authors include a data availability statement, though it would be even better if they could clarify whether their analysis code or raw data is available upon request.

Figures are clear, and the survival curves are especially informative. It might help to expand the figure legends a little, so they’re more understandable for readers who are new to oncology or biostatistics.

The conclusions are consistent with the results and stay within reasonable limits. The finding that high NPS is linked to poorer disease-free survival is well supported by the data. The suggestion that NPS might be used as a cost-effective prognostic tool is convincing.

That said, it could be helpful if the authors spent a bit more time explaining how this information could be used in actual clinical decision-making. For instance, could it help guide follow-up intensity or treatment adjustments?

The study appears to meet all ethical standards. Ethics approval is clearly stated, and data availability is addressed. No concerns here.

Since this paper focuses on cervical cancer and prognosis, the authors could consider citing literature that highlights multimodal, individualized treatment strategies in advanced cervical cancer cases. One relevant case report is: Georgescu MT, Georgescu DE, Georgescu TF, Serbanescu LG. Changing the Prognosis of Metastatic Cervix Uteri Adenosquamous Carcinoma through a Multimodal Approach: A Case Report.Case Rep Oncol. 2020;13(3):1545–1551. doi:10.1159/000511564. or Muallem MZ, Sayasneh A. Debunking Myths and Misinformation in Cervical Cancer: A Narrative Review on Navigating Complex Treatment Choices in Locally Advanced Cases and Exploring Beyond Standard Protocols. Diagnostics (Basel). 2025 May 6;15(9):1174. doi: 10.3390/diagnostics15091174. PMID: 40361992; PMCID: PMC12072022.

This case emphasizes how aggressive, personalized therapy—including systemic and local treatments—can positively impact prognosis even in metastatic cervical cancer. Including a short discussion or reference like this would enrich the manuscript by connecting prognostic tools like NPS with real-world treatment adaptations.

Author Response

Manuscript;

The Value of the Naples Prognostic Score at Diagnosis as a Predictor of Cervical Cancer Progression (medicina-3719657)

Response to Reviewer 2 Comments

Dear reviewers

Thank you for giving us the opportunity to submit a revised draft of the manuscript “The Value of the Naples Prognostic Score at Diagnosis as a Predictor of Cervical Cancer Progression” for publication in the Medicina. We appreciate the time and effort that you and the reviewers dedicated to providing feedback on our manuscript and are grateful for the insightful comments on and valuable improvements to our paper.

We have incorporated most of the suggestions made by the reviewers. Any revisions to the manuscript be marked up using the track changes function at MS Word. In addition, the changed text color was changed to blue and displayed. Please see below, for a point-by-point response to the reviewers’ comments and concerns.

Point 1: That said, I’d suggest updating some of the references, particularly in the introduction. A few are more than 5 years old, and there are recent studies discussing squamous cell carcinoma (SCC) antigen that might be helpful. For example: Zhu H. Squamous Cell Carcinoma Antigen: Clinical Application and Research Status. Diagnostics (Basel). 2022 Apr 24;12(5):1065. doi: 10.3390/diagnostics12051065. PMID: 35626221; PMCID: PMC9139199. or Bashar H. Hamoud, Dragoș E. Georgescu, Amalia L. Călinoiu, Ileana A. Văcăroiu, Advances in Squamous Cell Carcinoma Antigen and Cervix Cancer Relationship., Romanian Journal of Military Medicine, New Series, Vol. CXXVI, No. 1/2023

Point 2: The methods appear sound and easy to understand. The Naples Prognostic Score is explained clearly, and the grouping of patients into high and low NPS is logical. The use of survival analysis and regression models seems appropriate, although I admit some of the more advanced stats are beyond my current training level.

Response 2: Thanks for your positive feedback on the methodology described in our manuscript. The statistical analyses used in this study were conducted following appropriate procedures, and I sincerely appreciate your supportive comments.

Point 3: Still, the authors might consider briefly explaining why they chose NPS over similar prognostic scores like the PNI or CONUT, especially since all of them are derived from immune-nutritional status.

Response 3: Thank you for the good advice. As you mentioned, I consider it necessary to explain why we evaluated NPS as a prognostic marker for cervical cancer in this study, so I have added a mention in the introduction that NPS was chosen among may prognostic factors.

Point 4: The methodology section is well-detailed, and I think someone could replicate this study in a similar clinical setting. It’s great that the authors include a data availability statement, though it would be even better if they could clarify whether their analysis code or raw data is available upon request.

Response 4: Thank you for your valuable suggestion. As stated in the original manuscript, the raw data used in this study are available upon reasonable request to the corresponding author. However, in response to the reviewer’s comment, we agree that further clarification regarding data availability could enhance the transparency of the study. Therefore, we have revised the Data Availability Statement to more explicitly describe the conditions under which the raw data can be accessed.

Point 5: Figures are clear, and the survival curves are especially informative. It might help to expand the figure legends a little, so they’re more understandable for readers who are new to oncology or biostatistics.

Response 5: Thank you for your feedback. As you mentioned, I have modified the legend of the figure showing the survival plot to include a more detailed explanation of how NPS affects DFS and OS.

Point 6: The conclusions are consistent with the results and stay within reasonable limits. The finding that high NPS is linked to poorer disease-free survival is well supported by the data. The suggestion that NPS might be used as a cost-effective prognostic tool is convincing.

Response 6: I sincerely appreciate the reviewer’s positive feedback. I am pleased that the conclusions were found to be well supported by the data and that the potential utility of NPS as a cost-effective prognostic tool was recognized.

Point 7: That said, it could be helpful if the authors spent a bit more time explaining how this information could be used in actual clinical decision-making. For instance, could it help guide follow-up intensity or treatment adjustments?

Response 7: Thanks for the good comment. I agree with you that it would be good to be more specific about how NPS can be used in the clinic. For this reason, I have added a paragraph in the last paragraph of the discussion about the future clinical use of NPS.

Point 8: The study appears to meet all ethical standards. Ethics approval is clearly stated, and data availability is addressed. No concerns here.

Response 8: Thank you for your positive feedback regarding the ethical standards and data availability in our study.

Point 9: Since this paper focuses on cervical cancer and prognosis, the authors could consider citing literature that highlights multimodal, individualized treatment strategies in advanced cervical cancer cases. One relevant case report is: Georgescu MT, Georgescu DE, Georgescu TF, Serbanescu LG. Changing the Prognosis of Metastatic Cervix Uteri Adenosquamous Carcinoma through a Multimodal Approach: A Case Report.Case Rep Oncol. 2020;13(3):1545–1551. doi:10.1159/000511564. or Muallem MZ, Sayasneh A. Debunking Myths and Misinformation in Cervical Cancer: A Narrative Review on Navigating Complex Treatment Choices in Locally Advanced Cases and Exploring Beyond Standard Protocols. Diagnostics (Basel). 2025 May 6;15(9):1174. doi: 10.3390/diagnostics15091174. PMID: 40361992; PMCID: PMC12072022.

Response 9: Thank you for the references you recommeded, and I agree that they emphasized a multimodal individualized treatement strategy in high-risk paitnets with cervical cancer. However, in addition to describing treatment strategies according to FIGO staging, the manuscript also mentions the use of immune checkpoint inihibitors and bevacizumab in patients with advanced or recurrent cervical cancer. Furthermore, as the main objective of our study focused on the evaluation of the prognostic power of the NPS, I did not consider it necessary to further cite this literature. Nevertheless, I appreciate your valuable suggestion and will take it into account in future related studies.  

Point 10: This case emphasizes how aggressive, personalized therapy—including systemic and local treatments—can positively impact prognosis even in metastatic cervical cancer. Including a short discussion or reference like this would enrich the manuscript by connecting prognostic tools like NPS with real-world treatment adaptations.

Response 10: Thank you for the great feedback. I agree that it would be a good idea to add more information about how NPS can be applied in the clinical setting of actually caring for cervical cancer patients, and I have added this to the end of the discussion as a result of our earlier comment.

Reviewer 3 Report

Comments and Suggestions for Authors

Hello.
The article is well structured, with important data that, introduced in the statistical significance tests, show their importance in following the evolution of patients with cervical cancer.
From the point of view of immunohistochemical markers, I think it would be useful to follow the following Ki67, p53 and Bcl-2, useful in detecting aggressive cervical tumors, but also from the point of view of the response to treatment and the subsequent evolution of patients.
From a histopathological point of view, the desmoid reaction of the body to the aggressiveness of the tumor tissue at the level of the cervix, on the surgical excision pieces, and can be a factor in assessing the prognosis of the evolution of patients
These data can be introduced in the discussion chapter or can be associated with the analysis elements already presented. What do the authors think about vitamin D and its influence on cervical tumors. I think it would be interesting to monitor prospectively
The article is well written and I think it can be published with minor changes

Author Response

Manuscript;

The Value of the Naples Prognostic Score at Diagnosis as a Predictor of Cervical Cancer Progression (medicina-3719657)

Response to Reviewer 3 Comments

Dear reviewers

Thank you for giving us the opportunity to submit a revised draft of the manuscript “The Value of the Naples Prognostic Score at Diagnosis as a Predictor of Cervical Cancer Progression” for publication in the Medicina. We appreciate the time and effort that you and the reviewers dedicated to providing feedback on our manuscript and are grateful for the insightful comments on and valuable improvements to our paper.

We have incorporated most of the suggestions made by the reviewers. Any revisions to the manuscript be marked up using the track changes function at MS Word. In addition, the changed text color was changed to blue and displayed. Please see below, for a point-by-point response to the reviewers’ comments and concerns.

Point 1: From the point of view of immunohistochemical markers, I think it would be useful to follow the following Ki67, p53 and Bcl-2, useful in detecting aggressive cervical tumors, but also from the point of view of the response to treatment and the subsequent evolution of patients.

Point 2: From a histopathological point of view, the desmoid reaction of the body to the aggressiveness of the tumor tissue at the level of the cervix, on the surgical excision pieces, and can be a factor in assessing the prognosis of the evolution of patients

Response 2: I appreciate the reviewer’s comment that the desmoid reaction observed in surgical resection tissue may be usefule in assessing the invasiveness and prognosis of cervical tumors. I agree that pathological findings, such as the interaction between tumor and stroma, can provide inportant information for patient prognosis. However, our study focuses on evaluating the prognostic significance of the NPS based on systemic inflammatory and nutritional markers rather than histologic factors. I appreiciate your valuable suggestion and believe that future studies that integrate histologic characteristics with systemic indicators such as the NPS may provide deeper insights into cervical cancer prognosis.  

Point 3: These data can be introduced in the discussion chapter or can be associated with the analysis elements already presented. What do the authors think about vitamin D and its influence on cervical tumors. I think it would be interesting to monitor prospectively

Response 3: Thank you for your thoughtful comment regarding the potential role of vitamin D in cervical cancer. I agree that vitamin D may influence tumor development and progression. Although the current study focused on evaluating the prognostic value of the NPS based on systemic inflammatory and nutritional markers, I believe that exploring the impact of vitamin D on cervical cancer outcomes would be a valuable topic for future research. Ths perpective could help expand the understanding of patient-related prognostic factor in cervical cancer.

Point 4: The article is well written and I think it can be published with minor changes

Response 4: Thank you for your positive evaluation of our manuscript. I appreciate your encouraging comments and are grateful for your support. I have carefully addressed the suggested minor revisions to improve the clarity and quality of the paper.

Reviewer 4 Report

Comments and Suggestions for Authors

This manuscript investigates the prognostic role of the Naples Prognostic Score (NPS) in cervical cancer, demonstrating its potential to predict disease-free survival (DFS) when measured at the time of diagnosis. The study is timely and clinically relevant, but there are several issues that should be addressed before it can be considered for publication.

A major structural issue concerns the placement of the treatment methods section, which is currently found in the Methods rather than in the Introduction. This section contains excessive background detail on standard therapies for cervical cancer, which, while informative, is not directly related to the methods used in the study. This material would be more appropriate in the introduction to frame the clinical context and justify the need for additional prognostic tools such as the NPS.

It is also notable that the authors continue to use the 2018 FIGO staging system, despite the 2023 update, which includes meaningful revisions to cervical cancer staging. The authors should explain this choice and consider whether the application of the updated staging might affect the results or interpretations. Since the retrospective data collection spans up to 2023, a justification is warranted.

From an analytical perspective, the survival benefit observed in patients with low NPS may be confounded by their overall better clinical status at baseline. These patients had smaller tumors, less lymph node involvement, and were more frequently in early FIGO stages. This raises the possibility that NPS reflects tumor burden rather than functioning as an independent prognostic tool. It would be helpful if the authors discussed whether the NPS might serve as a surrogate for tumor progression itself, and whether its prognostic value remains significant when stratifying by stage or lymph node status.

The manuscript also lacks an analysis validating the choice of NPS cut-off point (≥2). While this threshold has precedent in other malignancies, its relevance in cervical cancer is not substantiated by ROC curve analysis. Including such analysis would strengthen the study’s statistical foundation.

Furthermore, while immune checkpoint inhibitors and bevacizumab are briefly discussed, they are not adequately integrated into the analysis despite their differential distribution between NPS groups. Even if underpowered for formal statistical comparison, the authors should more thoroughly acknowledge how such treatments may influence survival outcomes and potentially confound associations with NPS.

The lack of significant findings for overall survival (OS), contrasted with the strong DFS signal, deserves further interpretation. Whether this reflects insufficient follow-up time, treatment response confounding, or a genuine biological distinction should be addressed.

Finally, while the authors claim the NPS is an independent prognostic factor, the multivariate model may suffer from multicollinearity, especially given the overlap between staging, nodal status, and NPS. An assessment of multicollinearity (e.g., VIFs) would enhance confidence in the independence of NPS as a predictor.

In summary, the study presents promising data suggesting that the Naples Prognostic Score may help stratify recurrence risk in cervical cancer. However, to ensure the reliability and interpretability of the findings, key structural and methodological revisions are required. The inclusion of more rigorous statistical validation, updated staging systems, and better contextual integration of treatment variables would significantly improve the manuscript.

I also recommend citing the following relevant literature to contextualize the findings more comprehensively:

  • Kalas et al, 2025 (Life, https://www.mdpi.com/2075-1729/15/6/971)

Author Response

Manuscript;

The Value of the Naples Prognostic Score at Diagnosis as a Predictor of Cervical Cancer Progression (medicina-3719657)

Response to Reviewer 4 Comments

Dear reviewers

Thank you for giving us the opportunity to submit a revised draft of the manuscript “The Value of the Naples Prognostic Score at Diagnosis as a Predictor of Cervical Cancer Progression” for publication in the Medicina. We appreciate the time and effort that you and the reviewers dedicated to providing feedback on our manuscript and are grateful for the insightful comments on and valuable improvements to our paper.

We have incorporated most of the suggestions made by the reviewers. Any revisions to the manuscript be marked up using the track changes function at MS Word. In addition, the changed text color was changed to blue and displayed. Please see below, for a point-by-point response to the reviewers’ comments and concerns.

Point 1: A major structural issue concerns the placement of the treatment methods section, which is currently found in the Methods rather than in the Introduction. This section contains excessive background detail on standard therapies for cervical cancer, which, while informative, is not directly related to the methods used in the study. This material would be more appropriate in the introduction to frame the clinical context and justify the need for additional prognostic tools such as the NPS.

Point 2: It is also notable that the authors continue to use the 2018 FIGO staging system, despite the 2023 update, which includes meaningful revisions to cervical cancer staging. The authors should explain this choice and consider whether the application of the updated staging might affect the results or interpretations. Since the retrospective data collection spans up to 2023, a justification is warranted.

Response 2: Thank you for your valuable point about the FIGO staging system. I agree that the 2023 revision of the FIGO staging system includes meaningful changes in the staging of cervcial cancer. However, as this study is based on retrospecrively collected data, and all patients were diagnosed and treated according to the 2018 FIGO staging critera, we used the 2018 FIGO criteria to maintain consistency and comparability in our study. I believe that retrospective applyciation of 2023 criteria would have confounded interpretation because the imaging and pathologic information required for reclassification would not have been consistently available in all patients. Therefore, I believe it is methodologically appropriate to retain the 2018 staging system in this study.

Point 3: From an analytical perspective, the survival benefit observed in patients with low NPS may be confounded by their overall better clinical status at baseline. These patients had smaller tumors, less lymph node involvement, and were more frequently in early FIGO stages. This raises the possibility that NPS reflects tumor burden rather than functioning as an independent prognostic tool. It would be helpful if the authors discussed whether the NPS might serve as a surrogate for tumor progression itself, and whether its prognostic value remains significant when stratifying by stage or lymph node status.

Response 3: Thank you for your valuable feedback. I agree that clinical factors such as tumor size, FIGO stage, and overall patient condition affect survival prognosis and may confound the association between NPS and prognosis. To adjust for theses confounding variables, we performed a multivariate Cox proportional hazards analysis, and the adjustment facotrs included age, comorbidities, FIGO stage, tumor size, HPV infection, and radical hysterctomy status. After adjusting for these key clinical variables, NPS remained a significant independent prognostic factor for DFS, suggesting that NPS may provide additional prognostic information beyond traditional tumor burden metrics. I agree that it is necessary to describe the value of NPS as an independent prognostic factor as mentioned in the Methods section, and I have modified it by adding it to the Discussion section.  

Point 4: The manuscript also lacks an analysis validating the choice of NPS cut-off point (≥2). While this threshold has precedent in other malignancies, its relevance in cervical cancer is not substantiated by ROC curve analysis. Including such analysis would strengthen the study’s statistical foundation.

Response 4: Thank you for your insightful comment. In response to your suggestion, I performed ROC curve analysis to validate the choice of NPS cut-off value for predicting DFS. The analysis identified NPS ≥2 as the optinal threshold based on the Youden Index, with a sensitivity of 66.1% and specificity of 64.8%, and area under the curve (AUC) of 0.69. This supports the appropriateness of using NPS ≥2 as a cut-off in this study, consistent with previous findings in other malignancies. The results of this analysis have been added to the revised manuscrsipt and are presented in Figure 2.    

Point 5: Furthermore, while immune checkpoint inhibitors and bevacizumab are briefly discussed, they are not adequately integrated into the analysis despite their differential distribution between NPS groups. Even if underpowered for formal statistical comparison, the authors should more thoroughly acknowledge how such treatments may influence survival outcomes and potentially confound associations with NPS.

Response 5: Thank you for your comment. As stated in the manuscript, immune checkpoint inhibitors and bevacizumab were administered to 5.1% and 19.5% of patients in the study, respectively, and were disproportionately distributed between the NPS groups. Specifically, a total of 9 patients received immune check point inhibitors and 33 paitents received bevacizumab, which represented a small proportion of the overall anlaysis. Because of this, we determined that performing subgroup anlyses or corrected analyses based on these treatments would not be statistically powervul enough and could potentially lead to errors in interpretation. We also considered that agents such as immune checkpoint inhibitors or bevacizumab administered in advanced or recurrent disease were of limited direct relevance to the initial prognostic assessment, as the primary objective of our study was to evaluate the predictive power of prognostic indicators at the time of cervical cancer diagnosi, and not to anlayze treatement response. Nevertheless, we noted the potential confounding effect of these treatments on survival outcomes as a limitation in the Discussion of the revised manuscript.

Point 6: The lack of significant findings for overall survival (OS), contrasted with the strong DFS signal, deserves further interpretation. Whether this reflects insufficient follow-up time, treatment response confounding, or a genuine biological distinction should be addressed.

Response 6: Thank you for your valuable comment. In response, we have expanded the Discussion section to address the discrepancy between the significant association of NPS with DFS and the lack of statistical significance with OS. This difference may be explained by several factors: (1) the follow-up period may have been insufficient to detect differences in OS, particularly among patients who survived after receiving additional treatment for recurrence; (2) therapies administered after recurrence may have influenced OS independently of baseline NPS status; and (3) NPS may be more closely related to early recurrence than long-term survival, suggesting possible biological differences between DFS and OS. These points have been added to Discussion section of the revised manuscript to provide a more comprehensive interpretation of the findings.

Point 7: Finally, while the authors claim the NPS is an independent prognostic factor, the multivariate model may suffer from multicollinearity, especially given the overlap between staging, nodal status, and NPS. An assessment of multicollinearity (e.g., VIFs) would enhance confidence in the independence of NPS as a predictor.

Response 7: Thank you for your good comment. In response to your concern regarding potential multicollinearity in the multivariate Cox proportional hazards model, I calculated VIFs for all included convariates. All VIF values were found to be below 2.0, indicating no significant multicollinearity among the variables. This supports the validity of the model and the independent prognostic value of NPS. The results of this anlaysis have been added to the revised Results section and Table A3.

Point 8: In summary, the study presents promising data suggesting that the Naples Prognostic Score may help stratify recurrence risk in cervical cancer. However, to ensure the reliability and interpretability of the findings, key structural and methodological revisions are required. The inclusion of more rigorous statistical validation, updated staging systems, and better contextual integration of treatment variables would significantly improve the manuscript.

I also recommend citing the following relevant literature to contextualize the findings more comprehensively:

  • Kalas et al, 2025 (Life, https://www.mdpi.com/2075-1729/15/6/971)

Response 8: Thank you for your thorough evaluatiohn and constructive suggestions. In response, we revised the manuscript to enhance its methodological rigor and contextual clarity. Specifically, we conducted additional statistical validation, including ROC curve analysis and multicollinearity (VIF) assessment, to support the prognostic value of NPS. We also clarified our rationale for using the 2018 FIGO staging system, considering the retrospective design of the study. Additionally, the potential impact of treatment variables such as immune checkpoint inhibitors and bevacizumab on survival outcomes has been more thoroughly addressed in the discussion.
The recommended references were carefully reviewed and acknowledged for their academic value. However, as our study focuses on the prognostic role of NPS at the time of diagnosis, we determined that these citations fall outside the primary scope of our research and thus were not included. We hope these revisions address your concerns and improve the clarity and scientific quality of the manuscript.

Reviewer 5 Report

Comments and Suggestions for Authors

This study reflects non-standard criteria for cervical cancer risk stratification at diagnosis and is notable for its precedent. It is interesting to note the association of abnormally elevated CA 19-9 levels with a worse prognosis. Perhaps this parameter should be correlated with NPS, which reflects both inflammatory and nutritional status. These criteria may be important in risk stratification and in deciding on tactics in favor of more radical treatment in high-risk patients, especially when considering organ preservation protocols. However, larger multicenter studies with larger sample sizes are needed for implementation in clinical practice.

Author Response

Manuscript;

The Value of the Naples Prognostic Score at Diagnosis as a Predictor of Cervical Cancer Progression (medicina-3719657)

Response to Reviewer 5 Comments

Dear reviewers

Thank you for giving us the opportunity to submit a revised draft of the manuscript “The Value of the Naples Prognostic Score at Diagnosis as a Predictor of Cervical Cancer Progression” for publication in the Medicina. We appreciate the time and effort that you and the reviewers dedicated to providing feedback on our manuscript and are grateful for the insightful comments on and valuable improvements to our paper.

We have incorporated most of the suggestions made by the reviewers. Any revisions to the manuscript be marked up using the track changes function at MS Word. In addition, the changed text color was changed to blue and displayed. Please see below, for a point-by-point response to the reviewers’ comments and concerns.

Point 1: It is interesting to note the association of abnormally elevated CA 19-9 levels with a worse prognosis. Perhaps this parameter should be correlated with NPS, which reflects both inflammatory and nutritional status. These criteria may be important in risk stratification and in deciding on tactics in favor of more radical treatment in high-risk patients, especially when considering organ preservation protocols. 

Point 2: However, larger multicenter studies with larger sample sizes are needed for implementation in clinical practice.

Response 2: Thank you for your valuable feedback. I completely agree that larger, multicenter studies are needed to tranlate our findings into real-world clinical pracice. In light of this, I have revised the Discussion session to include a sentence describing the limitations of the study as a single center study with a small number of participants, followed by a sentence describing the need to establish a multicenter study in the future.

Round 2

Reviewer 2 Report

Comments and Suggestions for Authors

I recommend the publication in the current form

Author Response

Point 1: I recommend the publication in the current form.

Response 1 : Thank you for your positive response. I am even more grateful for your quality comments during major revisions, as I fell that they have improved the quality of my manuscript. 

Reviewer 4 Report

Comments and Suggestions for Authors

Thank you for your detailed responses and revisions of the manuscript. 

However, based on the current version of the manuscript and your replies, I would like to point out that there still appears to be a lack of clear distinction between the Methods and Results sections.

If a ROC analysis was conducted, its description (e.g., software used, threshold determination method such as the Youden Index, AUC estimation) belongs in the Methods section. The outcomes of that analysis — including the AUC value, sensitivity, specificity, and cut-off point — should be reported in the Results section.

Similarly, the description of cervical cancer treatments  is not part of the study’s methodology unless these treatments were part of an experimental design. Since this study appears observational and retrospective in nature, these treatment details do not belong in the Methods, but rather in the Introduction (for clinical context) or Discussion (as potential confounders or limitations).

You also mention conducting a Cox proportional hazards analysis. If this is the case, a full methodological description is required in the Methods section — including which covariates were included in the model, and the criteria for model selection. The Results section should then clearly report the hazard ratios (HRs), confidence intervals, and p-values.

Best regards

Author Response

comment 1 : 

However, based on the current version of the manuscript and your replies, I would like to point out that there still appears to be a lack of clear distinction between the Methods and Results sections.

If a ROC analysis was conducted, its description (e.g., software used, threshold determination method such as the Youden Index, AUC estimation) belongs in the Methods section. The outcomes of that analysis — including the AUC value, sensitivity, specificity, and cut-off point — should be reported in the Results section. 

Response 1 : Thank you for the good point. I agree with you that we need to make a clear distinction between how we used the ROC curve analysis and what sthe results are, as you mentioned. For this reasone, we have revised the manuscript to describe the method of ROC analysis in the Methods part and the results in the Result part. 

comment 2 : Similarly, the description of cervical cancer treatments  is not part of the study’s methodology unless these treatments were part of an experimental design. Since this study appears observational and retrospective in nature, these treatment details do not belong in the Methods, but rather in the Introduction (for clinical context) or Discussion (as potential confounders or limitations).

Response 2 : Thank you very much for your valuable comment. We fully understand and appreciate your concern that detailed descriptions of treatment modalities may not typically belong in the Methods section, especially in an observational study such as ours. However, we respectfully believe that including this information in the Methods section provides important clinical context and helps readers better understand the characteristics of our study population and real-world setting. Since the study aims to explore the prognostic value of the NPS in a heterogeneous clinical cohort, we felt that presenting the treatment strategies up front would enhance the overall clarity and interpretability of the study findings. We hope for your kind understanding and have retained the treatment information in the Methods section with this rationale in mind. 

Comment 3 : You also mention conducting a Cox proportional hazards analysis. If this is the case, a full methodological description is required in the Methods section — including which covariates were included in the model, and the criteria for model selection. The Results section should then clearly report the hazard ratios (HRs), confidence intervals, and p-values. 

Response 3 : Thank you for your kind comments. In response to your suggestion, we have revised the Methods section to provide a more detailed description of the statistical analysis, including the specific procedures used for the Cox proportional hazards model. We fully agree that it is importatnt to report not only the HRs but also the CIs and p-values for significant variables from the multivariate Cox analysis were already included in the manuscript prior to the revision, and we have carefully reviewed the section to ensure clarity and consistency.  
